# Effects of Biomass Fast Pyrolysis Fuel on the Tribological Behaviour of Heavy-Duty Diesel Engine Lubricating Oil

**Ruhong Song \*, Huiqiang Yu, Hui Song and Xianguo Hu**

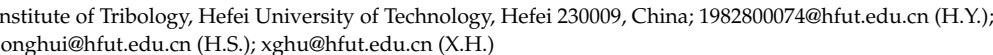

Institute of Tribology, Hefei University of Technology, Hefei 230009, China; 1982800074@hfut.edu.cn (H.Y.); songhui@hfut.edu.cn (H.S.); xghu@hfut.edu.cn (X.H.)
* Correspondence: 1984800043@hfut.edu.cn

**Abstract:** The fuel type not only influences the engine power and exhaust emissions, but dilutes the lubricating oil. We studied the effects of biomass fast pyrolysis fuel, or biofuel, on the tribological behaviour of a fully formulated engine oil (FFEO) used for heavy-duty diesel engines by reciprocating a sliding tribometer, which simulated the tribological conditions of an engine cylinder liner and piston ring. We analysed the surface morphology, surface roughness, and elemental contents of countersurfaces through scanning electron microscopy/energy dispersant spectroscopy and surface roughness measurements. The wear mechanism was studied by analysing the compositions and kinematic viscosities of the oil samples. The results indicate that the friction coefficient increased along with the emulsified biomass fuel (EBF) content in FFEO. The wear mass loss and EBF content were simultaneously increased. The wear mechanism was mainly attributed to the corrosion function of the biofuel.

**Keywords:** biofuel; lubricating oil; tribology; engine oil

## 1. Introduction

Biomass fast pyrolysis fuel is a renewable, environmentally friendly, and biodegradable fuel alternative that has attracted significant attention. This type of crude oil is produced through the fast thermal pyrolysis of biomass materials such as rice husks [1]. Using biofuel reduces vehicle emissions and subsequently improves fuel efficiency [2–7]. However, biomass crude fuels cannot be used directly in diesel engines because of their high water content, low thermal value, and high acidity. Emulsified biomass crude fuel and refined biomass fuel are the most efficient types of biomass fuel [8]. Regarding the use of emulsified biomass fuels in diesel engines, we should consider the effects of biofuel on the lubricating oil.

The tribological properties of biofuels have been studied extensively. Yang [9] affirmed that the waste derived from intermediate pyrolysis oils has better lubrication but could corrode certain parts of the engine. Lu [10] described the components of biomass crude oil, and the wear and friction mechanisms of biomass fuels were attributed to the organic chemical reaction of film formation on the friction pair under boundary lubrication conditions. The increase in wear during the rubbing process also increased the temperature, which resulted in corrosion. Xu [11] determined the tribological performances of biomass crude oil and refined biomass oil by using a four-ball tribometer and a high-frequency reciprocating test rig. The biomass crude oil lubricated better than the refined biomass oil because of the reduction in long-chain organic matter in the latter. The anti-friction properties of biomass fuels were better than those of diesel fuels. However, the anti-wear and anti-corrosion properties of biomass fuels were worse than those of diesel fuels.

The effects of these fuels on the tribological performance of engine oils have been examined. Masjuki [12] investigated how various percentages of palm oil diesel (POD)-contaminated lubricants affect the wear behaviour of cast irons during sliding contact. A

low percentage of POD (preferably 5% by weight) contamination functioned as an anti-wear additive that improved the performance of the lubricant on cast iron. Maleque [13] investigated the contamination of a lubricant with POD and the effects of a surface-hardening treatment on the wear and friction of AISI 1045 steel. Corrosive wears and pits were found on the specimen's surface when plain biofuel was used as a lubricant. This finding confirmed that the surface-hardened steel specimen had better wear resistance with a 4% biofuel-contaminated lubricant. Arumugam [14,15] reported that rapeseed-oil-based bio-lubricant and biodiesel-contaminated synthetic lubricant had better wear resistance and friction mitigation than the synthetic lubricant alone under similar operating conditions. Thus, long-term use of the newly formulated bio-lubricant and biodiesel might have a positive effect on engine life. The emulsified biomass fuel had no observable effects on the tribological behaviour of the engine lubricating oil between the cylinder liner and the piston ring friction pair.

Heavy-duty diesel is usually used in marine or tractor diesel engines, which are less fastidious than other diesel engines regarding lubricating oil. On the other hand, it is a little difficult to directly use biomass fuel in normal diesel engines because of the complex composition of biomass fast pyrolysis fuels. The successful use of biomass diesel will depend on solutions to many problems. One of aspects that should be focused on is the fuel's dilution of the lubricating oil in the engine, and we did exactly that in the present research. Thus, we studied the effects of the emulsified biomass fuel contents on the tribological performance of a fully formulated engine oil (FFEO, CD SAE 15W-40) in simulations of engine cylinder liners and piston rings. Due to the complex composition of the emulsified biomass fuel, the lubricating oil used in heavy-duty and tractor diesel engines (CD SAE 15W-40) was selected to act as the test oil. The experimental results provide some basis in data for the coming use of biomass fuels in diesel engines.

## 2. Experimental

### 2.1. Material and Apparatus

Both number zero diesel (DO) and the FFEO (CD SAE 15W-40) were purchased from Hefei Petrochemical Company. Crude biomass fuel (CBF), which was produced from rice husks via the fast pyrolysis process, was supplied by the Key Laboratory for Biomass Clean Energy in Anhui Province, China. Table 1 shows the composition of CBF as analysed via chromatography–mass spectroscopy. The analysis indicated that the crude biomass fuel could be classified as having four components: acid (considered as the main component), basic, non-polar, and polar. The other components included aldehyde, ketone, ester, alcohol, sugar, phenol, o-methylphenol, and furan [16].

**Table 1.** The compositional analysis of the crude biomass fuel by GC-MS.

| RT (min) | Area (wt%) | Composition |
|---|---|---|
| 1.42 | 3.14 | Formaldehyde |
| 1.51 | 6.52 | Acetaldehyde |
| 1.61 | 3.14 | Hydroxyl acetaldehyde |
| 1.72 | 2.70 | Hydroxyl acetone |
| 1.82 | 0.96 | Butyric acid |
| 2.07 | 29.78 | Acetic acid |
| 2.6 | 3.54 | Glyceraldehyde |
| 2.77 | 3.27 | 3,4-Dihydroxyl-di hydro-furan-2-ketone |
| 2.86 | 6.83 | 2,2-Dimethoxy ethanol |
| 3.13 | 6.56 | Furfural |
| 3.5 | 3.47 | 2,5-Dimethyl-4-tetrahydrofuran |
| 4.27 | 0.43 | 4-hydroxy-acid |

**Table 1.** *Cont.*

| RT (min) | Area (wt%) | Composition |
|---|---|---|
| 4.51 | 0.74 | 5H-furan-2-ketone |
| 4.76 | 1.31 | 2,3-Dimethyl-central ethanol |
| 5.19 | 0.38 | 3-Methyl-5H-furan-2-ketone |
| 6.15 | 1.18 | Maple lactone |
| 6.59 | 1.57 | Phenol |
| 6.8 | 1.12 | O-Methyl phenol |
| 7 | 1.46 | Cresol |
| 7.79 | 1.78 | 2,6-Dimethyl phenol |
| 8.99 | 1.14 | 3,4-Dimethyl phenol |
| 9.7 | 1.31 | 4-Ethyl-phenol |
| 10.1 | 1.53 | 3-(2-hydroxyl-pheyl) acrylic acid |
| 10.81 | 3.53 | Catechol |
| 11.9 | 1.36 | 3-Methyl catechol |
| 12.7 | 0.24 | Vanilin |
| 12.86 | 0.71 | 4-Ethyl catechol |
| 14.73 | 9.95 | Fructose |
| 15.5 | 0.20 | 2,3,4-Trimethoxy benzaldehyde |
| 15.8 | 0.15 | 3-(4-hydroxyl-2-methyl-pheyl) acrolein |

Table 2 presents the physical and chemical properties of the oil samples. Two surfactants (Span-80 and Tween-80) were purchased from Tianjin Guangfu Chemical Company. The cylinder liners and piston rings were purchased from Zhejiang Kaishan Cylinder Liner Company and Nanjin Feiyan Piston Ring Company, respectively. A 5 wt% emulsified biomass fuel (EBF) was prepared in accordance with the procedure shown in reference [17]. The experimental oil mixtures were prepared by adding 20, 40, 60, and 80 vol% EBF to the FFEO using a high shear emulsification machine (SG400 model) operating at 1500 rpm for 30 min. The other reagents, acetone and ethanol, were of analytical grade.

**Table 2.** The physical and chemical properties of oil samples.

| Items | DO | CBF | EBF | FFEO | Methods |
|---|---|---|---|---|---|
| C content (%, m/m) | 85.55 | 32.35 | 82.42 | - | GB/T 19143-2003 [a] |
| H content (%, m/m) | 13.49 | 8.36 | 13.23 | - | GB/T 19143-2003 [a] |
| O content (%, m/m) | 0.66 | 58.06 | 3.99 | - | GB/T 19143-2003 [a] |
| Heat Value (MJ·kg$^{-1}$) | 47.25 | 17.15 | 45.91 | - | ASTMD240 |
| Density (kg·m$^{-3}$, 20 °C) | 737.80 | 862.60 | 778.16 | 850 | ASTM D4052 |
| Kinematic viscosity (40 °C, mm$^2$·s$^{-1}$) | 2.71 | 2.10 | 3.09 | 110.6 | ASTM D445 |
| Solidifying point (°C) | −15 | <−58 | −14 | - | ASTM D5853 |
| Surface tensity (mN·m$^{-1}$, 20 °C) | 31.1 | 37.6 | 33.7 | - | ASTM D1417 |
| Acid number (mg·KOH·g$^{-1}$) | 0.12 | 33.10 | 0.82 | 0.035 | ASTM D664 |
| Water content(%, $v/v$) | Trace | 24.50 | 1.35 | Trace | ASTM D6304 |

[a] China National Measurement Standard.

## 2.2. Characterisation

The tribological experiments were conducted on an engine cylinder liner–piston ring tribometer (ECPT) assembled by the Institute of Tribology of Hefei University of Technology. The upper and lower cylinders were attached to the tester at a length of 122 mm, width of 15.6 mm, and height of 6.3 mm. The piston ring was 8 mm long and 2 mm wide. Although the tests were conducted at room temperature (approximately 25 °C), they had already been proven effective at achieving meaningful approximations of actual working engines [18–20]. Figure 1 presents a schematic of the ECPT. The oil samples were placed in an ultrasonic bath for 30 min before the tribological tests to improve the mixture of fuel and lubricating oils. The wear and friction properties of the oil samples were tested at a load of 210 N and a reciprocating frequency of 10 Hz for 6 h. The sliding distance of the piston ring was 80 mm and the speed of oil supply was 25 mL/h. The electrical signal denoting pressure

was obtained using a tension and compression sensor. The friction force was calculated by using the following formula [20]:

$$F = 20 \times U \tag{1}$$

$$M = F/L \tag{2}$$

$$\Delta m = m_1 - m_0 \tag{3}$$

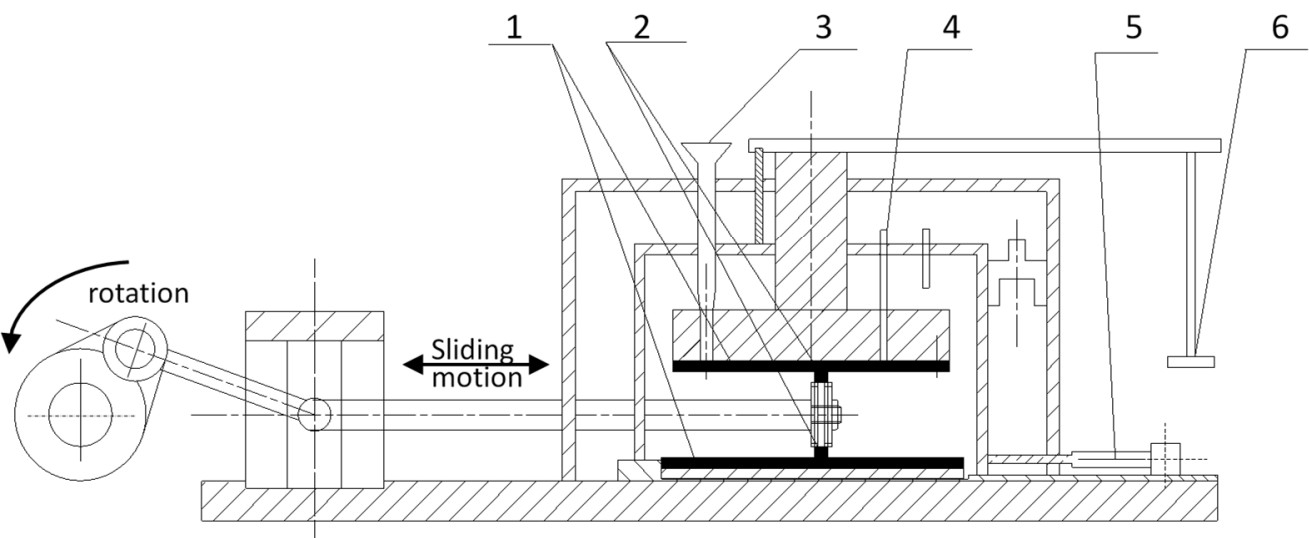

**Figure 1.** Schematic diagram of the tribometer simulating the friction pair of cylinder liner–piston ring. 1—cylinder liner, 2—piston ring, 3—fuel tank, 4—temperature sensor, 5—pressure sensor, 6—load.

SEM/EDS (JEOL Model JSM-6490) was used to investigate the morphologies and elements of the wear zones of the cylinder liner after the tribological tests. A surface roughness measurement system (Taylor-Hobson-6) was used to measure the roughness of the wear scars on the surfaces of cylinder liners that were magnified 10,000 times. The length of each sample was 0.08 mm, and the evaluation length was five times longer than the length of the sample, which was measured by using a Gaussian filter and sensor at 10,000:10 mm. The surface roughness is expressed by the contour arithmetic mean deviation *Ra*.

Fourier transform infrared spectroscopy (American Perkin Elmer Co., Waltham, MA, USA) was employed to investigate the active components of EBF and determine the effect of EBF on the wear mechanism of the FFEO. The kinematical viscosity of the oil samples was also investigated at 20, 40, and 60 °C. The wear mass loss was calculated based on Formula (3). The cylinder liner and piston ring were washed with acetone in an ultrasonic water bath for 10 min before the tribological tests to reduce experimental error.

## 3. Results and Discussion

### 3.1. Friction Reduction and Wear Resistance

Figure 2 reports the variations in friction coefficient among the oil samples at a load of 210 N and a reciprocating frequency of 10 Hz. The friction coefficients of all oil samples initially decreased for 45 min because of the running-in effect.

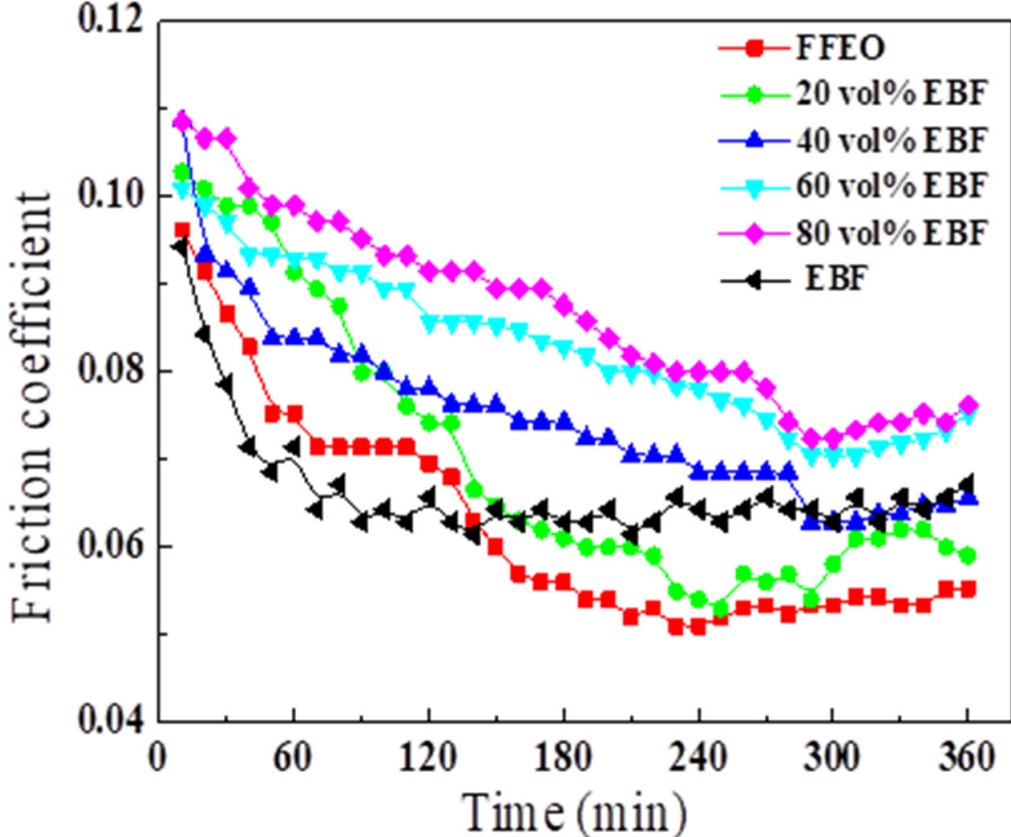

**Figure 2.** Variations in friction coefficient for various oil samples at a load of 210 N and a reciprocating frequency of 10 Hz.

The friction coefficients of the oil samples increased simultaneously with the EBF content, mainly because of the dilution function of the EBF. The thickness of the oil film generally increased along with the kinematic viscosity of oil. The trend of kinematic viscosity is discussed in Table 2. An increase in EBF content promoted corrosive wear. These results were verified by the degradation of the FFEO additives and decreases in EBF anti-friction components [21].

The friction coefficient of pure EBF was lower than that of pure FFEO during the 140 min initial rubbing process. Opposite results were observed after the rubbing process because of the easy absorption of active EBF components on the wear surface, forming a lubrication film during the initial rubbing process [14]. The low friction coefficient of FFEO was attributed to the antifriction and anti-wear additives in engine lubricating oil, whose mechanism will be discussed later.

Figure 3 presents the variations in the mass losses of the cylinder liner and piston ring with the different oil samples lubricated at a load of 210 N and at a reciprocating frequency of 10 Hz. The mass loss of the pure FFEO barely revealed the anti-wear property of the additives. The mass losses of the cylinder liner and piston ring increased after EBF content in the FFEO was increased. For the FFEO with low EBF contents (20 vol% and 40 vol%), the increases in mass loss suggest reduced efficiency of engine oil additives. With regard to the FFEO with high contents of EBF (60 vol% and 80 vol%), the increase in wear mass losses was also caused by the EBF's corrosiveness apart from the aforementioned reason. This phenomenon was explained by observing the surface of cylinder liners immersed in pure FFEO and EBF at 50 °C for 5 days. Obvious pit corrosion caves, as shown in Figure 4c, were found on the surfaces of cylinder liners immersed in EBF [22].

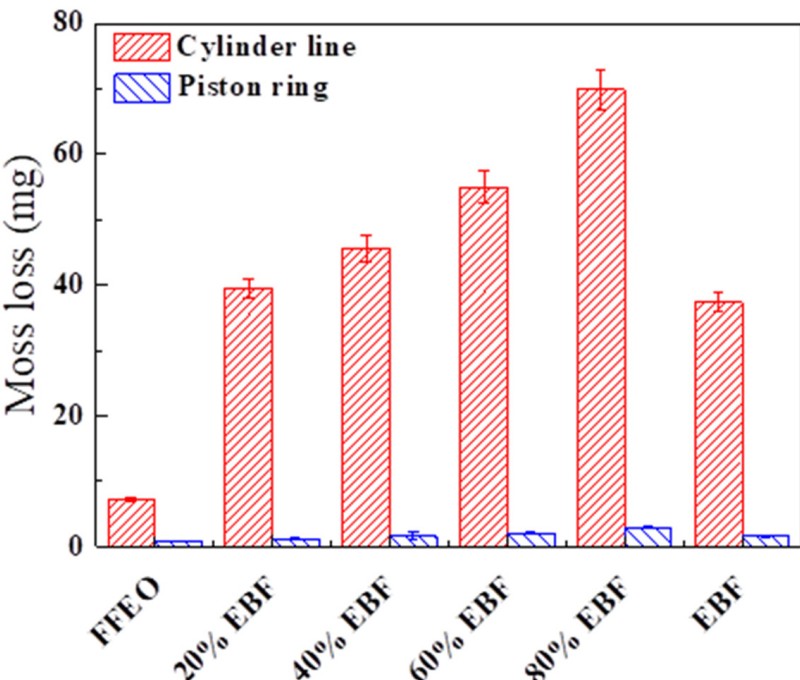

**Figure 3.** Variations in mass losses of the cylinder liner and piston ring with different oils.

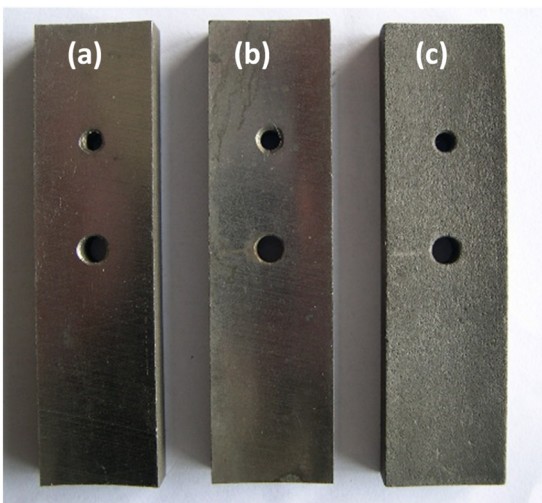

**Figure 4.** The surface diagrams of cylinder liners immersed in FFEO and EBF oils at 50 °C for 5 days. (**a**) Initial, (**b**) FFEO, (**c**) EBF.

The mass losses of the cylinder liner and piston ring in pure EBF were less than those of samples in EBF-contaminated oil, but were higher than those of samples in pure FFEO. These results indicate that the anti-wear property of EBF is worse than that of FFEO, and provide a plausible explanation for why the anti-wear property was degraded as the EBF content in the FFEO increased. The lubrication component of EBF was consumed by its interaction with additives in the FFEO [23]. With 80 vol% EBF, the maximum mass losses of the cylinder liner and piston ring were as much as 69.9 and 2.9 mg. This phenomenon was caused by the interaction of EBF with FFEO and the corrosive wear of EBF.

### 3.2. Surface Analysis

Figure 5 shows the SEM images of the wear zones in the cylinder liners of different oil samples. Figure 5a presents the initial surface morphologies of the cylinder liners. The preparation process produced numerous caves, as indicated by the white arrows in the

SEM images. Figure 5b illustrates the surface morphology of the cylinder liner after it was tested with pure EBF.

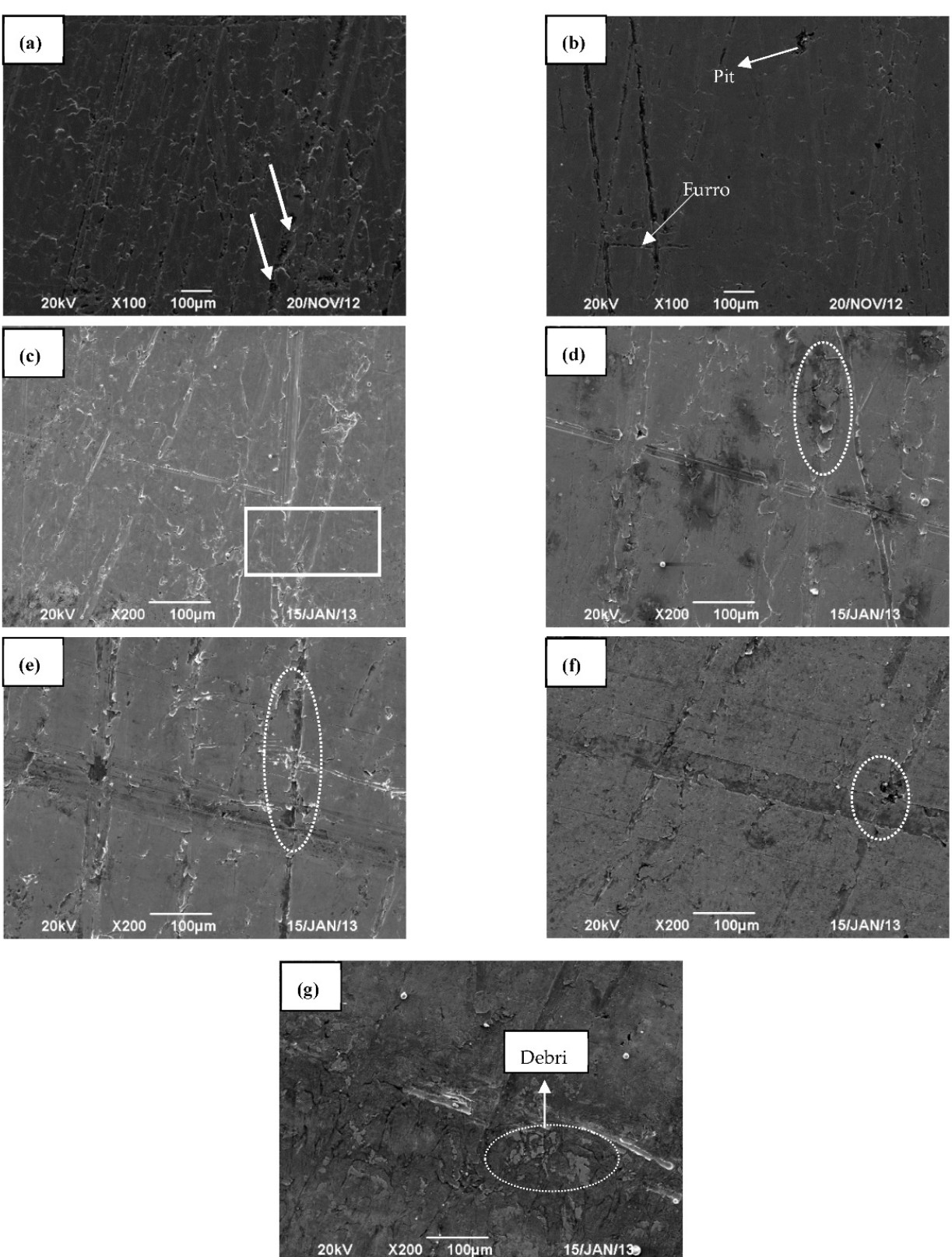

**Figure 5.** SEM images of the wear zones in cylinder liners. (**a**) Initial, (**b**) EBF, (**c**) FFEO, (**d**) 20 vol% EBF, (**e**) 40 vol% EBF, (**f**) 60 vol% EBF, (**g**) 80 vol% EBF.

Machining marks, corrosion pits, and furrows were observed on the surfaces of the cylinder liners. The corrosion pits were attributed to the acid and oxidised matter in the EBF. Figure 5c shows that the smooth surface was produced by the FFEO lubrication. The machining marks on the surface of the wear zone also confirmed that the FFEO provided better lubrication than the EBF. The peeling of debris could have been due to the fatigue wear with 20 vol% EBF in the FFEO. However, Figure 5d also indicates a smooth surface. After the EBF content was increased to 40 vol%, plenty of debris was found on the surface of the wear zone of the cylinder liner, which was produced by the EBF's corrosiveness, as illustrated in Figure 5e. On the one hand, the metal surface softened because of the corrosiveness of the EBF component. On the other hand, the soft-state matrices were easily peeled off during the rubbing process [24].

Figure 5f illustrates the corrosion phenomenon. Both the machining marks and furrows became lighter, and numerous pits disappeared compared with the case using 20 vol% and 40 vol% EBF in FFEO. This phenomenon was attributed to the corrosiveness of EBF. More wear debris and crackles were found on the surface of the cylinder liner exposed to 80 vol% EBF, as shown in Figure 5g. These observations would explain the corrosiveness of EBF, which was consistent with the results presented in Figure 4c.

Figure 6 shows the variations in contour profile and surface roughness of cylinder liners lubricated by different oils. There are many parameters to describe a surface contour profile, such as *Ra*, *Rsm*, *Rsr*, and *Rz*, which depend on different definitions of surface contour profile. We selected *Ra*, the arithmetic mean of the absolute value of contour offset within the sampling length, to characterize the surface roughness of cylinder liners lubricated by different oils. In actual measurements, the more measuring points, the more accurate *Ra* is.

Figure 6a–c indicates that the cylinder liner lubricated with pure FFEO had higher surface roughness than the liners lubricated with initial and pure EBF. This was due to the properties of the different oil samples. Generally, oil with high kinematic viscosity results in a thick layer of the better-lubricating oil film. The initial crackles were protected by the FFEO oil film during the rubbing process. The low surface roughness of pure EBF was attributed to the corrosiveness of the biomass oil and the thinner oil film compared with that of FFEO, as shown in Figure 6b [25]. The surface roughness increased and then decreased when 20, 40, and 60 vol% EBF were added to the FFEO, as shown in Figure 6d–f, because of the negative effect of EBF on the film thickness of FFEO.

The surface roughness ($R_a$ = 0.419) of the sample lubricated with 80 vol% EBF in the engine lubricating oil, as shown in Figure 6g, was close to that of the sample lubricated with pure EBF ($R_a$ = 0.417). This result, which is also indicated in Figure 5g, confirmed the corrosion function of EBF.

X-ray dispersive energy (EDS) was employed to investigate the elements and their contents in wear zones lubricated with 20 or 80 vol% EBF in the FFEO. Figure 7a and Table 3 (Lubri [a]) show the wear zone of the cylinder liner lubricated with 20 vol% EBF in FFEO, as determined via energy spectroscopy and elemental analysis. Zinc, calcium, and magnesium were detected and attributed to the additives in the FFEO, which would indicate that the lubrication film's formation was mainly influenced by the additives. On the other hand, it is important to control the EBF content in engine lubricating oil during the application of EBF in the engine, because the chemical composition, depth, mechanical properties, bonding strength with substrate, and ability to form a tribofilm depend on EBF in the FFEO.

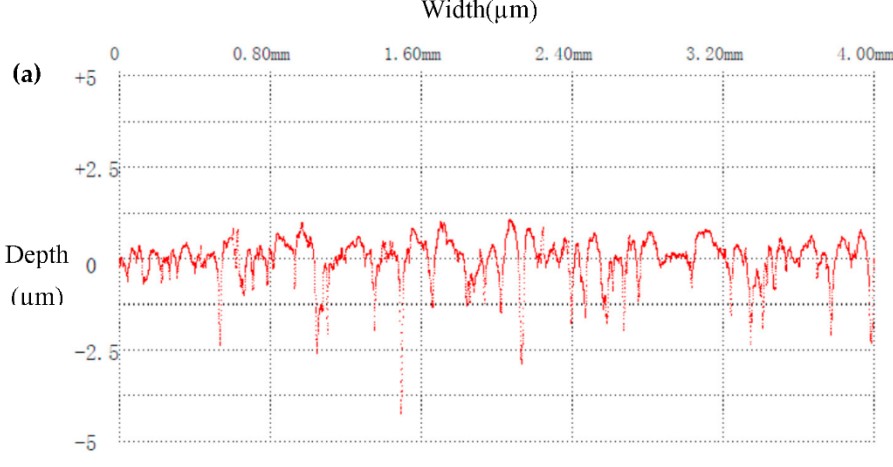

**Ra= 0.442 μm**

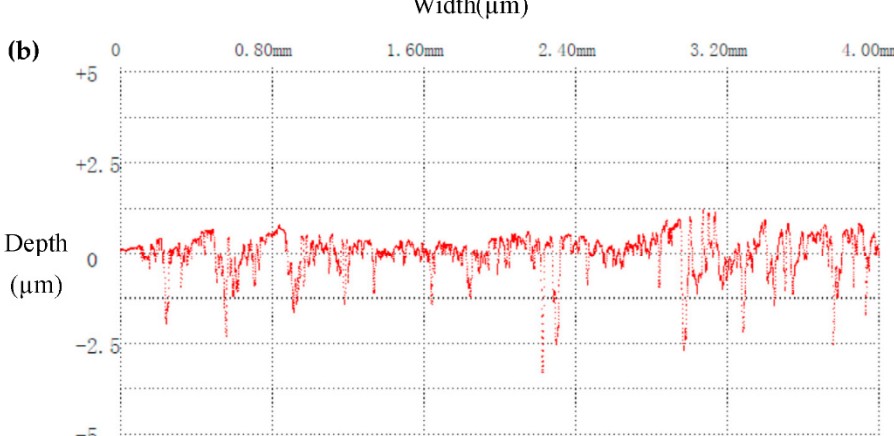

**Ra =0.419 μm**

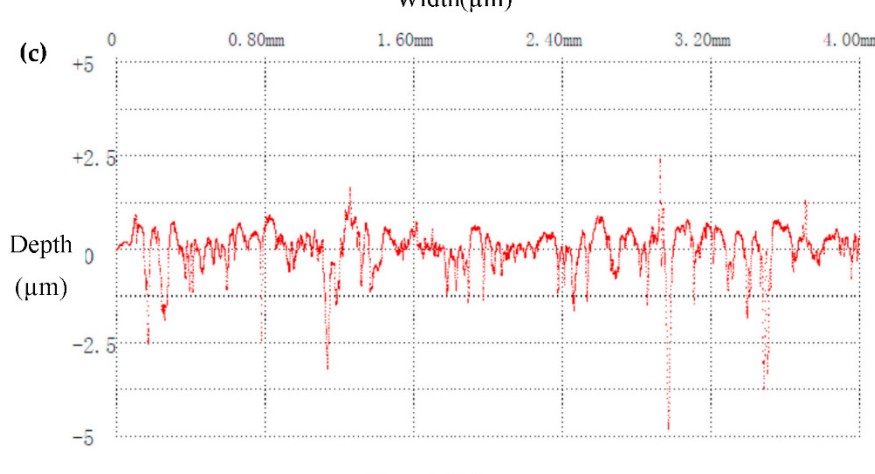

**Ra = 0.451 μm**

**Figure 6.** *Cont.*

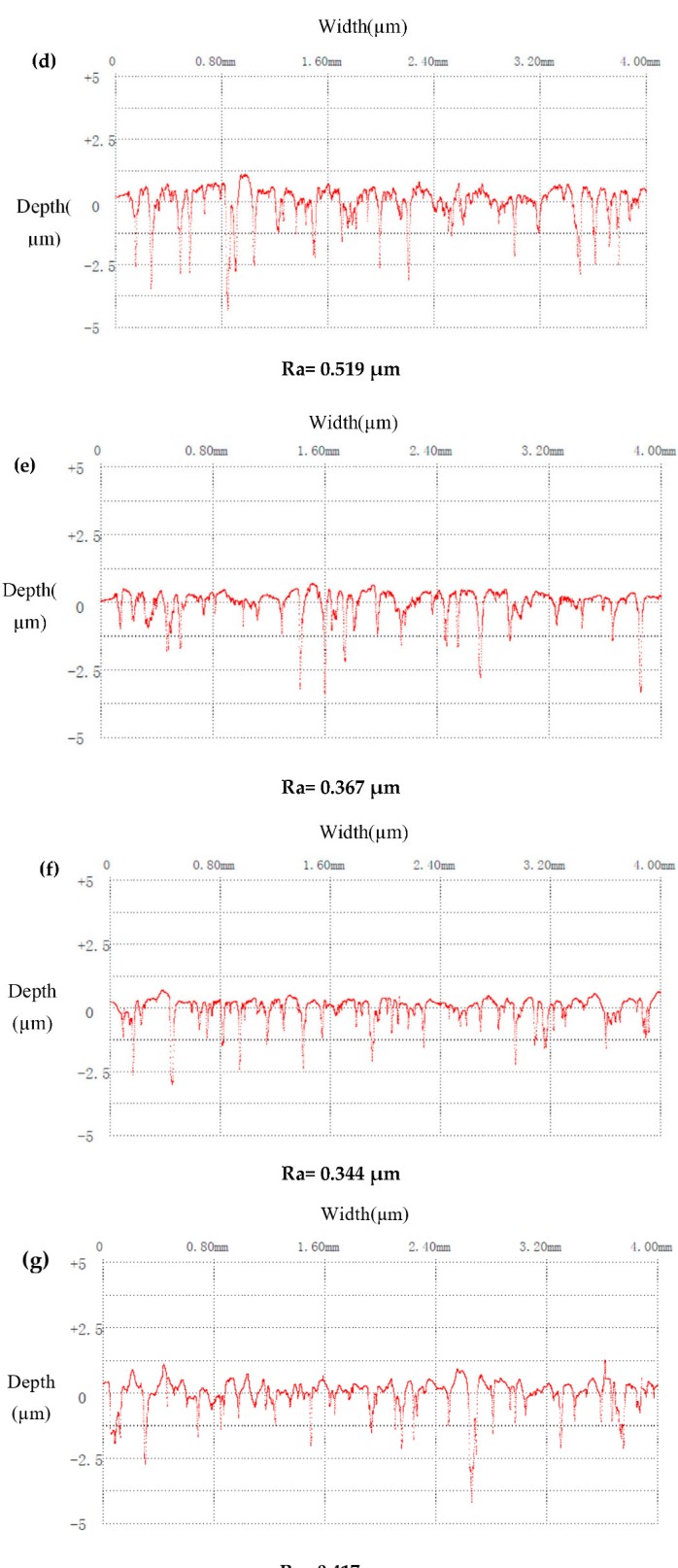

**Figure 6.** Variations in the contour profiles and surface roughness of cylinder liners lubricated by different oils. (**a**) Initial, (**b**) FFEO, (**c**) EBF, (**d**) 20 vol% EBF, (**e**) 40 vol% EBF, (**f**) 60 vol% EBF, (**g**) 80 vol% EBF.

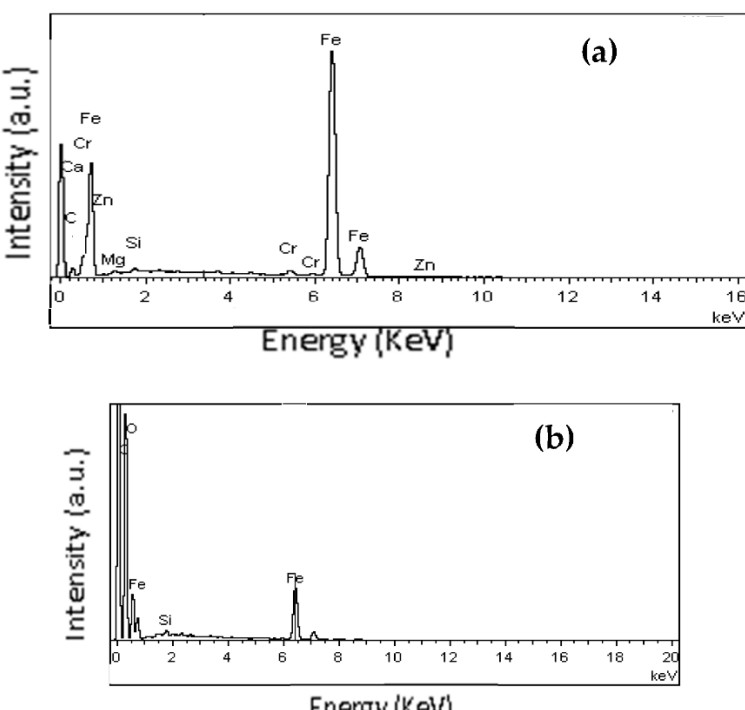

**Figure 7.** EDS analysis of cylinder liner wear zones with 20 and 80 vol% EBF in FFEO: (**a**) 20 vol% EBF, (**b**) 80 vol% EBF.

**Table 3.** EDS analysis of the cylinder liner wear zones lubricated with 20 vol% and 80 vol% EBF in FFEO.

| Item | Element Content (wt%) | | | | | | | |
|---|---|---|---|---|---|---|---|---|
| | **C** | **O** | **Fe** | **Zn** | **Cr** | **Mg** | **Si** | **Ca** |
| Lubri [a] | 0.91 | - | 95.88 | 0.22 | 1.15 | 0.81 | 0.77 | 0.26 |
| Lubri [b] | 22.74 | 31.86 | 43.12 | - | - | - | 2.29 | - |

[a] 20 vol% EBF, [b] 80 vol% EBF.

Figure 7b and Table 3 (Lubri [b]) illustrate the wear zone of the cylinder liner lubricated with 80 vol% EBF in FFEO, as determined via energy spectroscopy and elemental analysis. Zinc, calcium, and magnesium were not detected, which indicates that the FFEO additives did not contribute to the lubrication film formation. This result is consistent with the findings shown in Figure 5g. The EBF prevented lubrication film formation. Its corrosiveness also inhibited lubrication film formation. Higher levels of oxygen and carbon were detected on the surface, which confirmed the occurrence of oxidised or corrosion reactions [26].

*3.3. Wear Mechanism Analysis*

The wear mechanism of the biomass fast pyrolysis fuel on the piston ring and cylinder liner was investigated via composition analysis and varying the kinematic viscosity of the oil samples. Figure 8 shows FT-IR spectra of EBF and FFEO, in which EBF shows more active function groups from 5 wt% crude biomass oil compared with FFEO. The broad peak range at 3000 to 3600 cm$^{-1}$ was attributed to the free hydroxyl (-OH) in the crude biomass oil. Peaks at 1751 and 1615 cm$^{-1}$ were detected in two oil samples, which indicated the existence of acids, ethers, and esters. The main functional groups shown in Figure 8 are the same as those found in other literature [27], even though these biomass oils could have been produced from different raw materials or processes; their compositions were also very complex and variable. These active function groups would influence the wear mechanism, because they would react with the FFEO additives to decrease the oil film thickness during the rubbing process. The thinner oil film resulted in serious wear that could explain the

decrease in the surface roughness when the EBF content was as much as 60 vol%. The oil film thickness was also attributed to the variations in kinematic viscosity.

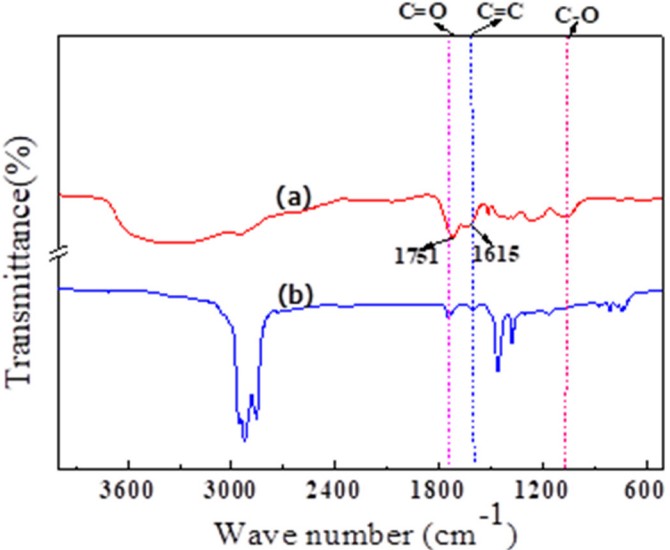

**Figure 8.** FT-IR analysis of biomass crude oil and emulsified biomass fuel: (**a**) biomass crude oil, (**b**) emulsified biomass fuel.

Figure 9 illustrates the variations in kinematic viscosity of the oil samples as a function of temperature. The viscosity decreased as temperature and/or EBF content in FFEO increased. The effects of corrosive wear also had a key function in promoting wear during the rubbing process. This observation is illustrated in Figures 4c and 5g.

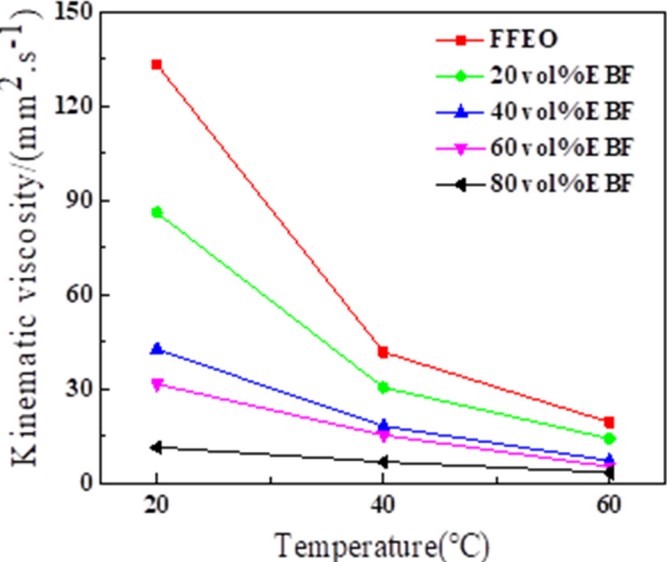

**Figure 9.** Variations in kinematical viscosity of oil samples as functions of temperature.

The surface asperities were easily polished, which resulted in low surface roughness, high friction coefficient, and increased mass loss when EBF was added to FFEO. The simple schematics in Figure 10 explain the effect of EBF on the tribological behaviour of FFEO.

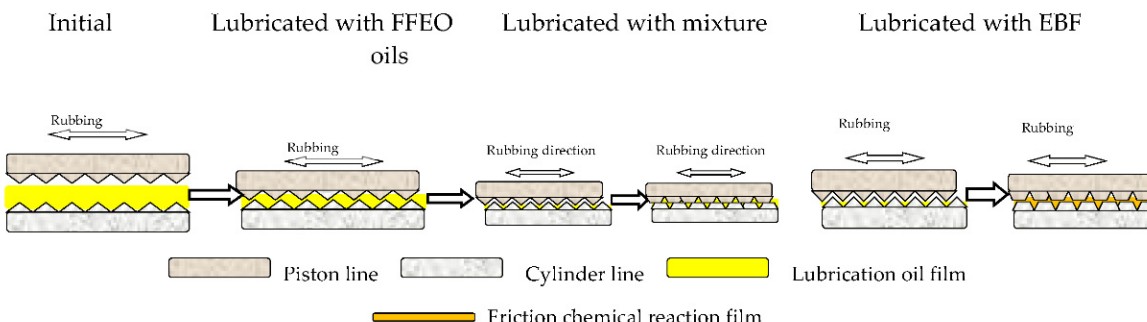

**Figure 10.** Schematic of the wear behaviours of cylinder liners and piston rings lubricated with different oils.

The surface asperities of the cylinder liner and piston ring were protected by the lubrication oil film through the FFEO. As shown in Figure 10, the polishing of surface asperities and decrease in oil film thickness when the EBF was added to the FFEO subsequently decreased the surface roughness of the cylinder liner during the rubbing process. The increase in surface roughness upon addition of as much as 80 vol% EBF was attributed to the corrosion function of EBF.

The active components of pure EBF were easily absorbed on the frictional pair to form a tribochemical film, which could explain the low friction coefficient and mass losses of the cylinder liner and piston ring, especially for the tribo-condition in the state of boundary lubrication. These findings will be helpful to understanding the tribochemical roles of compositions that include EBF together with the engine lubricating additives and basestocks.

## 4. Conclusions

The following conclusions were drawn by studying the effects of biomass fast pyrolysis fuel on the tribological behaviour of heavy-duty diesel engine lubricating oil with the help of a reciprocating sliding tribometer simulating the tribological conditions between a cylinder liner and piston ring. The friction coefficients and wear mass losses of the cylinder liner and piston ring increased simultaneously with the emulsified biomass fuel content in the fully formulated engine oil. The wear mass loss of the cylinder liner was 24 times larger than that of the piston ring when the emulsified biomass fuel content was as much as 80 vol% in the fully formulated engine oil. At the same time, the fully formulated engine oil was degraded by the emulsified biomass fuel, as reflected in the decrease of kinematic viscosity and the degradation of additives. The wear mechanisms of the cylinder liner and piston ring lubricated by fully formulated engine oil that was diluted by emulsified biomass fuel were attributed mainly to the corrosiveness of the biomass fast pyrolysis fuel.

**Author Contributions:** Conceptualization, R.S.; methodology, H.Y. and H.S.; validation, R.S.; investigation, H.Y. and R.S.; resources, H.S.; writing—original draft preparation, R.S.; writing—review and editing, X.H. All authors have read and agreed to the published version of the manuscript.

**Funding:** This research received no external funding.

**Informed Consent Statement:** Informed consent was obtained from all subjects involved in the study.

**Acknowledgments:** The authors wish to express their thanks to YuFu Xu for his helpful discussion about the experimental work.

**Conflicts of Interest:** The authors declare no conflict of interest.

## Abbreviations

| | |
|---|---|
| FFEO | Fully Formulated Engine Oil |
| EBF | Emulsified Biomass Fuel |
| DO | Number zero diesel |
| CBF | Crude Biomass Fuel |
| ECPT | Engine Cylinder Liner-Piston Ring Tribometer |
| SEM/EDS | Scanning Electron Microscope/Energy Dispersive Spectrometer |
| FT-IR | Fourier Transform InfraRed Spectrometer |
| F | Friction force (N) |
| U | Voltage (V) |
| L | Load (N) |
| $\Delta m$ | Wear mass loss (mg) |
| $m_0$ | Initial mass of the cylinder liner and piston ring (mg) |
| $m_1$ | Mass of the cylinder liner and piston ring after the rubbing process (mg) |
| Ra | Surface roughness ($\mu$m) |

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
