# Peer review of "Effects of Biomass Fast Pyrolysis Fuel on the Tribological Behaviour of Heavy-Duty Diesel Engine Lubricating Oil"

_applsci, doi:10.3390/app12052360_

Round 1
Reviewer 1 Report
The subject is interesting and actual.
It will be of interest to the readers.
Major amendments are needed:
1) the abstract is poor. Rewrite and present the general idea, the methodology and major achievements in 200 words.
2) The figures are representative but the quality is poor. Must follow the template of the journal. Improve this point.
3) Conclusions are poor. Must rewrite resuming the results and achievements of the work.
Author Response
Response to Reviewer 1 Comments
Point 1: the abstract is poor. Rewrite and present the general idea, the methodology and major achievements in 200 words
Response 1: Good ideas! The new abstract (133 words) contained the esteemed suggestions .
Point 2: The figures are representative but the quality is poor. Must follow the template of the journal. Improve this point.
Response 2: Some figures were improved.
Point 3: Conclusions are poor. Must rewrite resuming the results and achievements of the work.
Response 3: The Conclusions were rewrotten.
Reviewer 2 Report
1-Abstract: Adjust the spacing of your manuscript also briefly specify the importance of this research.
2-In the introduction, please specify the gap of knowledge and the novelty of your research, aims, and objectives.
3-The methodology is well written. It specifies the adopted approach and ensures the repeatability of the experiment and it is well referenced. However, please define the abbreviations before using them.
4-Please characterize Figure 6 statistically using the mean values and the stranded deviation. Also, make sure that the figure appears only on one page.
5-enhance the resolution of Figure 7.
6-Please reposition all figures and tables so that they are parallel to the text.
7-Compare the conclusion of Figure 8 to other research papers.
8-Figure 10 shall be referred to in the methodology section
9- The conclusion is not accepted as points. Please rewrite this section, please follow the adopted approach in other papers in MDPI.
10-Please add a discussion section, at the moment your paper appears as a report. You need to properly highlight the importance of your finding with extended.
11-Add nomenclature table.
12-Add a list of abbreviations
Author Response
Response to Reviewer 2 Comments
Point 1: Abstract: Adjust the spacing of your manuscript also briefly specify the importance of this research.
Response 1: The abstract by modified the importance of this research was also presented briefly.
Point 2: In the introduction, please specify the gap of knowledge and the novelty of your research, aims, and objectives.
Response 2: The aim and goal of the current research were also shown in the section of introduction.
“One of aspects should be focused on the fuel dilution role on the lubricating oil in engine, which is the aim and goal of the present research.”
Point 3: The methodology is well written. It specifies the adopted approach and ensures the repeatability of the experiment and it is well referenced. However, please define the abbreviations before using them.
Response 3: The related abbreviations have been shown on a separarted page by a list of abbreviations.
Point 4: Please characterize Figure 6 statistically using the mean values and the stranded deviation. Also, make sure that the figure appears only on one page.
Response 4: Good suggestions. Firstly, the surfacer roughness of worn surfaces were the typical tribological parameters. To express the surfacer roughness accuratulely is very completed research which is not the present research focus topic. The corrent reseafrch only show that different oils will lead varied surface roughness .
In addition, “figure appears only on one page” depends on the final version of manuscript.
Point 5: enhance the resolution of Figure 7.
Response 5: The appearance of Figure 7 was from the recorder diractly, thus the resolution of Figure 7 was low, but those findings in Figure 7 were shown clearly.
Point 6: Please reposition all figures and tables so that they are parallel to the text.
Response 6: Some Figures were replaced, and depended on the journal arrangements finally.
Point 7: Compare the conclusion of Figure 8 to other research papers.
Response 7: The additives in enigne lubricating oil were very complex, their compositions were also different from varied addive manufacrurers. The spectra of main function groips were same with other literatures.
Point 8: Figure 10 shall be referred to in the methodology section
Response 8: In genreal, the expalination of wear mechanism, like schamatic diagram of wear mechanism, was placed after the experimental results in some tribological journals, like Tribological International, Wear etc. Of course, itr is also accepted for Figure 10 being referred to in the methodology section.
Point 9: The conclusion is not accepted as points. Please rewrite this section, please follow the adopted approach in other papers in MDPI.
Response 9: The conclusions was rewroteted based on the esteemed suggestions and comments.
Point 10: Please add a discussion section, at the moment your paper appears as a report. You need to properly highlight the importance of your finding with extended.
Response 10: It was stressed the importance of our findings in the disscussion after some interesting results. Such as: “On the other hand, it is important to control the EBF content in engine lubricating oil during the application of EBF in the engine.” “These findings will be helpful to understand the tribochemical role of compositions of EBF together with the engine lubricating additives and basestocks.” etc.
Point 11: Add nomenclature table.
Response 11: A nomenclature table was added on a separated page.
Point 12: Add a list of abbreviations
Response 12: A list of abbreviations was added on a separated page.

Reviewer 3 Report
The manuscript is about the Effect of Biomass Fast Pyrolysis Fuel on the Tribological Behaviour of Heavy-duty Diesel Engine Lubricating Oil
All figures are not properly described.
As for the characteristics of biomass, the literature review was limited, the authors should take into account similar studies that have been carried out in the discussion. The authors described the subject of biomass in a limited way. I recommend that you refer to biomass in more comparisons of chemical aspects, I recommend reading the article: https://doi.org/10.3390/pr9020364 .For example in the above article, the PY-GC-MS of straw was shown.
It would be also good to describe the economic impact of the technology used and compare how much the commonly used technology (an example estimate by the used fertilizers) would cost to that proposed by the authors' technology. This is important because the cost of implementing the technology is the basis for its application.
Author Response
Response to Reviewer 3 Comments
Point 1: All figures are not properly described.
Response 1: Most of figures are improved and properly described.
Point 2: As for the characteristics of biomass, the literature review was limited, the authors should take into account similar studies that have been carried out in the discussion. The authors described the subject of biomass in a limited way. I recommend that you refer to biomass in more comparisons of chemical aspects, I recommend reading the article: https://doi.org/10.3390/pr9020364 . For example in the above article, the PY-GC-MS of straw was shown.
Response 2: The ner version of manuscript cited the refence mentioned (https://doi.org/10.3390/pr9020364) and disscussied it.
Point 3: It would be also good to describe the economic impact of the technology used and compare how much the commonly used technology (an example estimate by the used fertilizers) would cost to that proposed by the authors' technology. This is important because the cost of implementing the technology is the basis for its application.
Response 3: The current research is a fundamental topic considering biomass fast pyrolysis fuel used in engine. It will be focused on the economic aspect in the future.

Round 2
Reviewer 1 Report
The authors addressed all my questions. I recommend to be published.
Reviewer 2 Report
Comments 4-8 and 10 from the previous review are not resolved.
A major correction is still required.
Reviewer 3 Report
The authors prepared the manuscript correctly to the comments. There is only a lack of proper editing.
Round 3
Reviewer 2 Report
The authors have addressed the major comments. The remaining comments can be improved in the proofreading stage.
(Overall Evaluation: Accept in present form)